# Family attitudes, actions, decisions and experiences following implementation of deemed consent and the Human Transplantation (Wales) Act 2013: mixed-method study protocol

Jane Noyes,[1] Karen Morgan,[2] Phillip Walton,[3] Abigail Roberts,[4] Leah Mclaughlin,[1] Michael Stephens[5]

For numbered affiliations see end of article.

**Correspondence to**
Jane Noyes;
jane.noyes@bangor.ac.uk

## ABSTRACT

**Introduction** The Human Transplantation (Wales) Act 2013 (the Act) introduced a 'soft opt-out' system of organ donation on 1 December 2015. Citizens are encouraged to make their organ donation decision known during their lifetime. In order to work, the Act and media campaign need to create a context, whereby organ donation becomes the norm, and create a mechanism for people to behave as intended (formally register their decision; consider appointing a representative; convey their donation decision to their families and friends or do nothing—deemed consent). In addition, family members/appointed representatives need to be able to put their own views aside to support the decision of their loved one. The aim of this study is to evaluate initial implementation, outcomes and impact on families and appointed representatives who were approached about organ donation during the first 18 months.

**Methods and analysis** Prospective mixed-method coproductive study undertaken with National Health Service Blood and Transplant (NHSBT), and multiple patient/public representatives. The study is designed to collect information on all cases who meet specified criteria (≥18 years, deceased person voluntarily resident in Wales and died in Wales or England) whose family were approached between 1 December 2015 and 31 June 2017). Data for analysis include: NHSBT routinely collected anonymised audit data on all cases; Specialist Nurse in Organ Donation (SNOD) completed anonymised form for all cases documenting their perception of the families' understanding of the Act, media campaign and outcome of the donation approach; questionnaires and depth interviews with any family member or appointed representative (minimum 50 cases). Additional focus groups and interviews with SNODs. Anonymised donation outcomes and registration activity reports for Wales provide additional context.

**Ethics and dissemination** Approved by NHSBT Research, Innovation and Technology Advisory Group on 23 October 2015; Wales Research Ethics Committee 5 (IRAS190066; Rec Reference 15/WA/0414) on 25 November 2015 and NHSBT R&D Committee (NHSBT ID: AP-15–02) on 24 November 2015.

## Strengths and limitations of this study

► The study is a large-scale prospective mixed-method evaluation of the immediate impact of the Act using multiple data sources.
► Previous studies have struggled to recruit family members involved in an organ donation approach.
► The success of this study is dependent on the multiple recruitment strategies and the engagement of NHS Blood and Transplant staff who will primarily recruit participants.
► NHS Blood and Transplant is an equal partner in this coproductive study.
► Many patient and public representatives and organisations are supporting the study.
► CRUSE Bereavement Care Cymru is supplying bereavement support information for bereaved families who participate in the study.

**Registration** The protocol is registered on the Health and Care Research Wales Clinical Research Portfolio. Study ID number 34396, www.ukctg.nihr.ac.uk

## INTRODUCTION

The Human Transplantation (Wales) Act 2013 introduced a 'soft opt-out' system of organ donation.[1] In an 'opt-out' system presumed consent means that unless the deceased person has expressed a wish in life *not* to be an organ donor then consent will be assumed (or deemed in Wales). There are two types of 'opt-out' system: a 'hard opt-out' where the family is not consulted or a 'soft opt-out' where the family is consulted.[2]

The purpose of the Act is to make it easier for people to donate their organs to benefit patients. The Act is central to the Wales Action Plan,[3] which sets out a programme of continuous improvement on all aspects of organ donation and transplantation to

deliver the NHS Blood and Transplant (NHSBT) strategy 'Taking Organ Transplantation to 2020'.[4] NHSBT is a Special Health Authority in England and Wales (accountable to the Department of Health) that is responsible for promoting tissue and organ donation to the public and managing organ donation and transplantation. The overall target of the strategy is to increase the UK consent rates to 80% by 2020. Under the former 'opt-in' system, in 2012/2013, 2013/2014 and 2014/2015 only 50.3%, 53.6% and 48.5% of families consented to deceased donation in Wales, respectively.[5] In contrast, the consent rate in Spain, which operates an 'opt-out' system in which all citizens are automatically registered for organ donation unless they choose to state otherwise, ranged between 80% and 85%.[4 6]

Wales has a devolved parliamentary legislature within the UK and a population of just over 3 million people. Responsibility for healthcare legislation is devolved to the Welsh Government. The Human Transplantation (Wales) Act 2013 constitutes one of the biggest changes to the partnership and social contract between the Welsh Government and the people of Wales. The Act is however controversial and not everyone consulted agreed with the 'soft opt-out' system and its principle of deemed consent.[7 8]

Potential donor families are considered to be most affected by the Act as, unlike the old 'opt-in' system, their role in the 'soft opt-out' system remains essential but changed by deemed consent.[9] Under the previous 'opt-in' system, which came under the Human Tissue Act 2004,[10] if the individual's consent had not been indicated by the deceased person or a nominated representative, consent was sought from the person who was in a 'qualifying relationship' with the deceased person immediately before their death (usually a family member). If the decision regarding donation was unknown then families were less likely to give consent.[9 11] If those close to the deceased person objected to organ donation, for whatever purpose, when the deceased person (or their nominated representative) had explicitly consented, they did not have the legal right to revoke the consent, however the existence of appropriate, valid consent permitted donation to proceed, but did not mandate that it must. The final decision about whether to proceed rested with the medical team when family members did not support donation.

## HOW THE INTERVENTION IS INTENDED TO WORK?
In a research context, the Act and implementation strategy is conceptualised as a complex behaviour change intervention.[12] The Act changes the principles of consent to deceased organ donation from one of 'opt-in' to a 'soft opt-out' for adults who are 18 years or over; voluntarily resident for 12 months or more in Wales; who have not made an express decision regarding organ donation and is competent to understand the notion of deemed consent. The individual must also die in Wales for the Act to apply.

NHSBT employ teams of Specialist Nurses in Organ Donation (SNODs), who work across regions to support the organ donation process.[13] The choices individuals now have in either expressing their organ donation decision or choosing to do nothing and having their consent deemed (criteria apply) have impacted on the approach to the family by the SNODs. Once the SNODs have ascertained that the individual has not recorded their organ donation decision on the Organ Donor Register (ODR) and has not appointed a representative to make the decision on their behalf, the conversation with the family is presumptive in favour of organ donation, informing them if applicable their relatives' consent will be deemed to have been given. During the conversation, the family is able to inform the SNODs that their relative did not want to be an organ donor. In this circumstance, the family is required to produce clear evidence that the person did not want to be an organ donor. The Act is permissive in the sense that it allows for consent to be deemed in certain circumstances, however, it does not mandate that organ donation goes ahead in such cases. If an individual has registered a decision or informed someone that they did not want to donate organs prior to their death, their decision will be respected unless the family is able to produce clear evidence that the individual had changed his/her mind.

## INTENDED BEHAVIOUR CHANGE
The success of the Act depends on behaviour change (public and professional) to work as intended. The theory is that the neutral media campaigns supporting implementation will facilitate five behaviours:
1. People will register to 'opt-in' on the organ donor register and appoint a patient representative;
2. they register to 'opt-out';
3. people will discuss their donation decision with families and friends;
4. people can do nothing and it will be assumed that they do not object to organ donation (deemed consent);
5. in making the donation decision, families will put aside their own views on donation and respect the decision of the deceased person.

Overall, this complex intervention addresses four components of behaviour change as outlined in the Nuffield Council of Bioethics ladder of intervention (figure 1).[14] The Act and implementation strategy were designed to change the default position so that organ donation became the norm. The Government-led media campaign was however presented in a neutral way to provide people with information to make an informed choice. Nudge theory was also used to underpin behaviour change—such as exposing the population of Wales to a series of 'nudge alerts' via email, Royal Mail and the media to do specific things such as making their organ donation decision known and 'opting in' or 'out' on the organ donation register.[14] The media did however

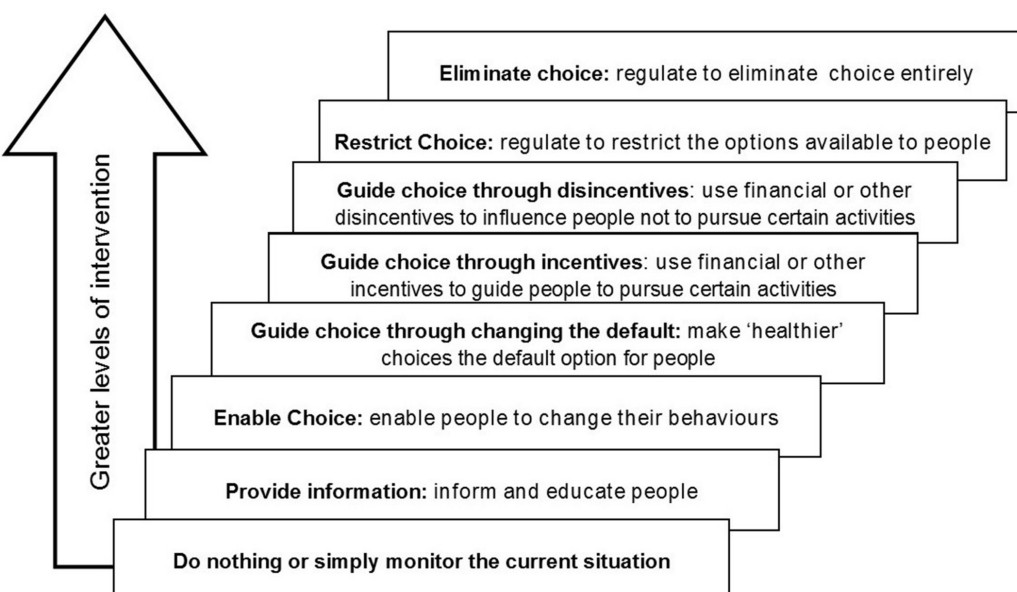

**Figure 1** Nuffield council of bioethics ladder of intervention.

generally present organ donation as having positive benefits (eg, giving the gift of life).

In addition to the public media campaign, there was an accompanying implementation strategy for NHS and NHSBT staff, which required amending clinical protocols and procedures and retraining large numbers of staff and all SNODs covering Wales. The multiple elements of this complex intervention are shown visually in figures 2, 3 and 4.

## MODIFICATIONS TO THE APPROACH CONVERSATION UNDER THE ACT

The SNOD facilitates an approach conversation with the family at the point indicated in figure 5.

After 1 December 2015, for deceased people who have not registered to 'opt-out' on the ODR, the approach to families will be a presumptive conversation in favour of organ donation. The sequence of obtaining consent for deceased organ donation when the patient has not recorded their decision on the ODR is shown in figure 6. Irrespective of whether the deceased person is registered on the ODR or not, the assumption is that family members will put aside their own beliefs if different to the deceased person and support the express decision to donate or by choosing not to register a decision by any means support their relative's deemed consent.

## PRUDENT HEALTHCARE PRINCIPLES

The Act is conceived as a Prudent healthcare policy. Any Prudent health service or intervention is based on the following four principles[15 16]:

► Achieve health and well-being with the public, patients and professionals as equal partners through coproduction. Patient and public contribution is essential to create a patient-centred system for both potential donors and transplant recipients.[15 16] The soft opt-out system has been developed in close consultation with the people of Wales.[17–20]

► Care for those with the greatest health need first, making the most effective use of all skills and resources. The principles underpinning organ transplantation decisions are founded on caring for those with the greatest health need first, irrespective of ability to pay. There is good evidence that all transplants are cost-effective. For example, the cost benefit of kidney transplantation compared with dialysis over a period of 10 years (the median transplant survival time) is £241 000 or £24 100 per year for each year that the patient has a functioning transplanted kidney.[21] Although the Act covers all organs and tissues from which patients may benefit from cost-effective transplants, the case for economic renewal and regeneration is best made in Wales by increasing the number of kidney transplants. Kidney transplants are highly cost-effective particularly in relation to NHS spend, and is the treatment of choice for many patients with end-stage renal failure. Recipients can often engage more productively in the economy once they no longer need dialysis.

► Do only what is needed, no more, no less; and do no harm. The 'soft opt-out' is designed to make it easier for the people of Wales to become organ donors. Transplantation is designed to offer patients more options for their treatment with increased benefits that outweigh the risks.

► Reduce inappropriate variation using evidence-based practices consistently and transparently. Attitudes to organ donation vary across Wales and across social gradients and cultures.[20] The purpose of the neutral media campaign is to reduce this variation by providing the public with high-quality accessible information.

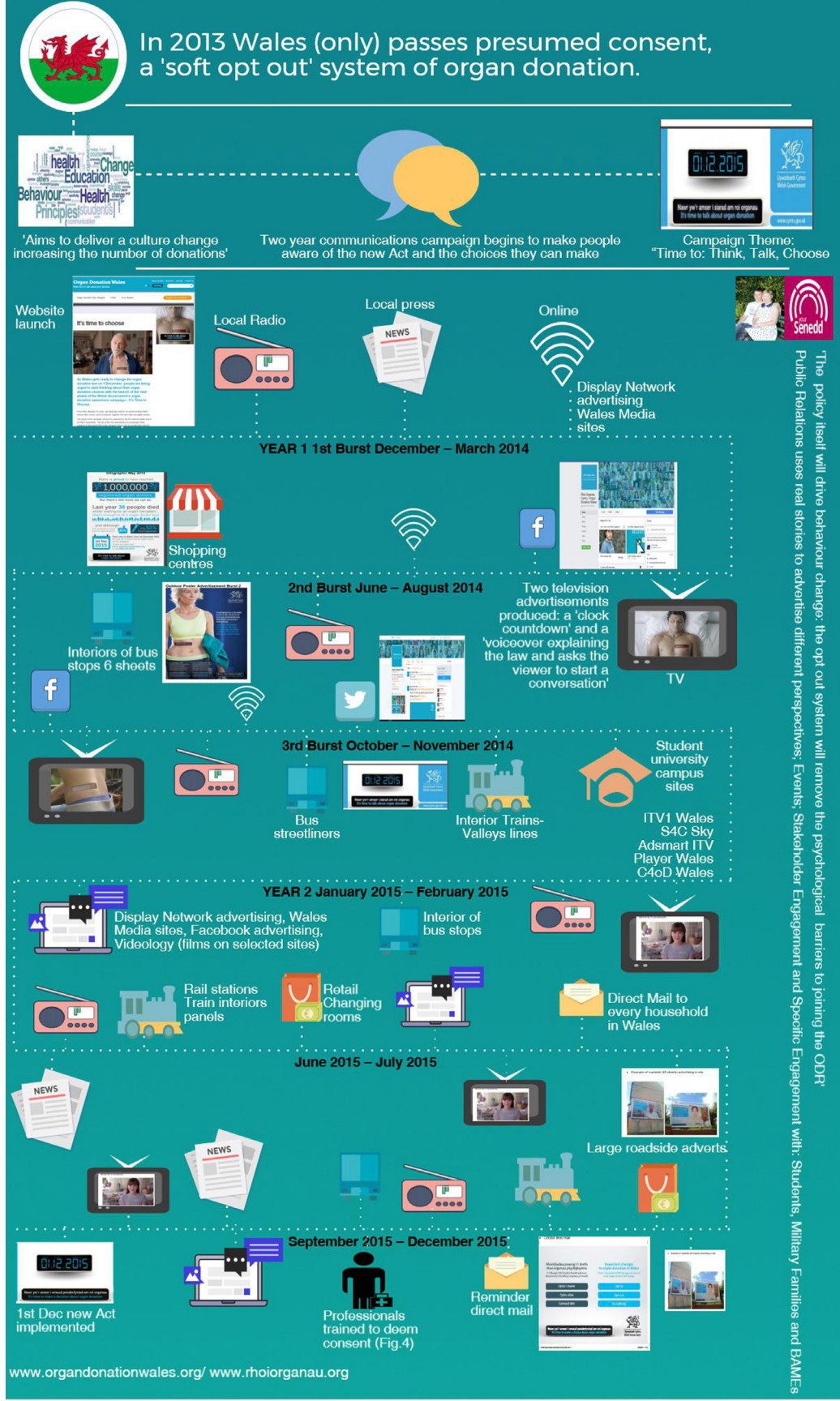

**Figure 2** Intervention implementation: multifaceted media-based strategy to inform the public of the Act and changes to consenting to organ donation.

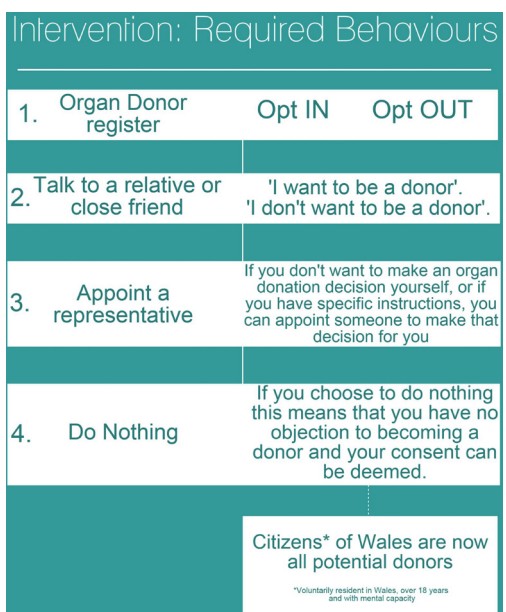

**Figure 3** Intervention and required behaviours following introduction of the Act.

## RATIONALE FOR THE STUDY

There is evidence from a UK context describing the multiple converging factors that appear to influence donation decisions under the 'opt-in' system, such as knowledge of the deceased's wishes and the view of families that the deceased person had suffered enough.[15–20] We want to specifically explore the perspectives of organ donor registration and deemed consent with families and close friends who were involved in an organ donation decision.

This study is designed to address a critical gap in understanding by exploring if the Act has changed the views and decisions of families. The research is needed to understand donor family responses, which could have an immediate impact on the design of future interventions to change behaviours. Understanding how and why people in reality respond to the 'soft opt-out' will be vital to contextualising the impact of this Prudent health policy in achieving its aims. We want to explore what happened and ascertain the perspectives and decisions made by individuals who were involved in an organ donation decision, and to explore whether the donation decision reflected the patient or family view. There is also a potential benefit to participants as the study provides a confidential independent opportunity to talk about their views and experiences, which in turn can be used to benefit future donor families and patients.

Findings will fill a critical gap in knowledge to supplement the Welsh Government impact evaluation and shed light on the mechanisms that prevent or enhance organ and tissue donation under the new 'soft opt-out' system. Undertaking research to better understand these mechanisms and how they work will be vital for policy makers, healthcare professionals working in NHSBT and the NHS in general. It will inform continuous service improvement to realise the intended outcomes of this very complex intervention (the Act, media-based behaviour change interventions, retraining of NHS and NHSBT staff and the interventions of NHS and NHSBT teams when requesting consent).

## AIM

The aim of the study is to explore the impact of the Act on consent for deceased organ and tissue donation in the new 'soft opt-out' system. A secondary aim is to further build research capacity in NHSBT and Patient and Public Involvement (PPI) representatives in Wales.

## RESEARCH QUESTIONS

1. What impact and changes has the Act and media campaign had on the views and decisions of families of potential organ donors in Wales?
2. What were the views of the deceased person and how did families take account of the deceased person's view in the decision-making process?
3. What are the views of families of the deceased person on the shift in relationship with the Government and healthcare services; organ donor registration; deemed consent; express patient decision and role of appointed representatives and the changed role of families in decision making in a 'soft opt-out' system?

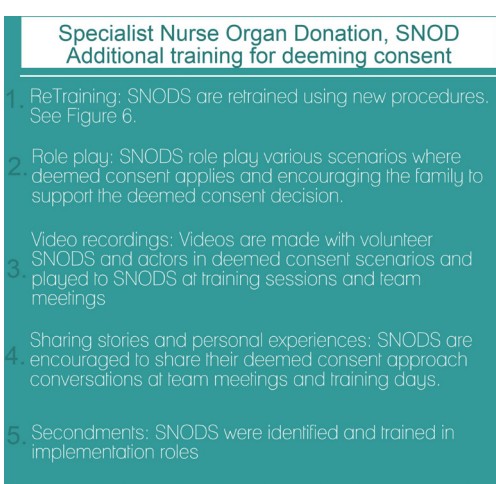

**Figure 4** Intervention implementation: additional training for Specialist Nurses in Organ Donation (SNOD) when approaching families following the implementation of deemed consent.

## OBJECTIVES

1. To ascertain a broad overview of anonymised family views, actions and outcomes from organ donation conversations in Wales for an 18-month period following implementation of the Act.
2. To explore in greater depth the perspectives and experiences of families who were involved in a donation conversation.

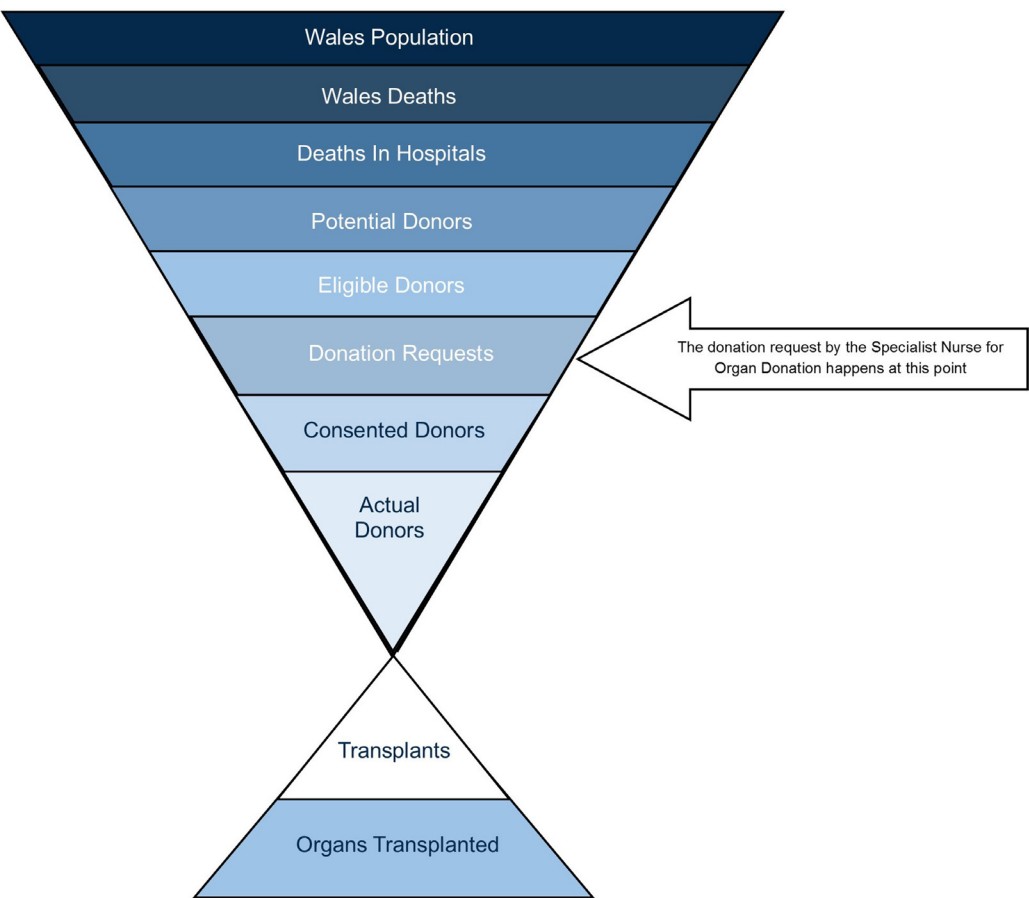

**Figure 5** Wales potential organ donor population and identification of the 'donation request' stage in the process.

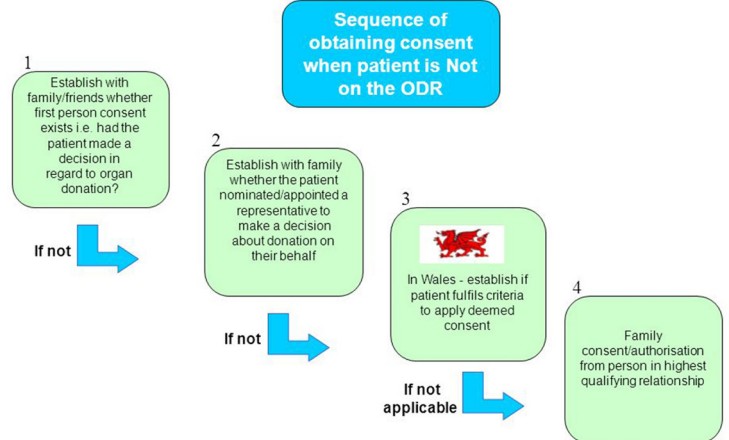

**Figure 6** Sequence of obtaining consent when the patient is NOT on the Organ Donor Register (ODR).

3. To explore the perspectives of SNODs and their managers covering Wales to contextualise potential donor family views, experiences and decisions.
4. To contextualise findings with Welsh Government survey data and contemporaneous and previous NHSBT activity reports on organ donor registration and organ donation in Wales.
5. To further develop research capacity and capability in NHSBT and patient and public representatives in Wales.

## METHODS AND ANALYSIS
We consulted widely and extensively with multiple key stakeholders to design an ethically defensible and sensitive study that respects the vulnerability and confidentiality of bereaved potential donor families and the dignity of the deceased family member. The four-phase design (figure 7) combines use of routinely collected donor audit activity and national attitudinal surveys as context to a primary study using shared anonymised and routinely collected NHSBT information on decision-making

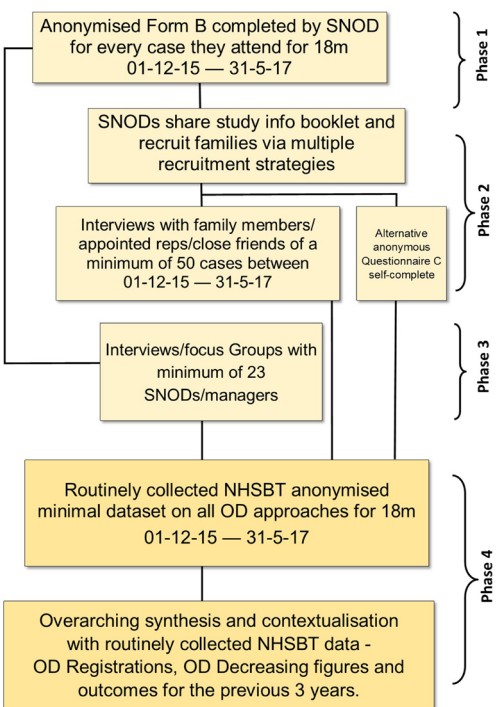

**Figure 7** The four-phase study design. NHSBT, National Health Service Blood and Transplant; OD, organ donation; SNOD, Specialist Nurse in Organ Donation.

processes and outcomes of the donation consent process, and interviews with Welsh potential deceased donor families/appointed representatives/close friends and organ donation professionals covering Wales (see online supplementary file 1) for a summary of all data sources contributing to the analysis). Recruitment and data capture targets for each phase are shown in figure 7.

Phase I: primary study to gain a high level understanding of the impact on donor family responses (accepting the patient's decision, or consenting to, or not consenting to donation) for 18 months from 1 December 2015.

SNODs will complete an anonymised electronic one page form (see online supplementary file 2): form B for every approach conversation that will be filled out as soon as possible after they have disengaged from speaking with the potential donor family. No participant or patient identifiable information will be recorded. SNODs will complete the form (electronic or paper) using information gathered from their routine conversations with potential donor families.

Phase II: primary study with potential deceased donor families/appointed representatives and close friends to ascertain a deeper understanding of their thoughts, experiences and responses to the Act and their decision making.

Family members/appointed representatives and close friends, directly or indirectly involved in the donation process, will be invited to self-complete an anonymised questionnaire (see online supplementary file 3): form C that requires no contact with the research team. There is no restriction on the number of questionnaires per family

and the questionnaire will take around 20 min to complete. Accompanying the questionnaire will be an invitation to participate in an interview to discuss their views and experiences in greater depth and a contact form to send back to the independent research team to arrange a mutually convenient interview (see online supplementary file 4 family interview schedule). For those wanting also to participate in an interview, several options will be offered that best suit the individual, such as face to face, telephone or via social media. Mindful that participants have been bereaved, they can select the time that is right for them to be interviewed up until the end of the period of data collection.

### Recruitment of family/close friend/appointed representative

We will use a range of methods that are sensitive and individually tailored to recruit participants involved in a minimum of 50 potential organ donation cases, with maximum variation to cover all donation pathways and outcomes. SNODs will use their discretion and knowledge of the family to select the most appropriate options and times to share information about the study with families/appointed representatives. Recruitment options include via direct contact with families by SNODs (with the option to using consent to contact form), and sharing study information in person; and by sending out a study invitation with information attached to routine follow-up communication by NHSBT; by direct mailing of study invitation and information by NHSBT; via adverts in the media, and through snowball sampling.

If an individual receives more than one letter of invitation, we will include a sentence to explain that, if they have already made their decision whether to participate in the study or not, they can ignore the letter or pass the invitation onto another family member or close friend of the deceased person, because NHSBT only have one contact name for each family. For participants who would prefer to be interviewed in their first language (Welsh or other language), we have employed a Welsh medium research officer and have built-in interpreter costs.

### Inclusion criteria for 'family' participant recruitment
► Any person over 16 years with mental capacity who was involved, either directly or indirectly, in a deceased organ donation conversation or decision in Wales after 1 December 2015.
► Any person over 16 years with mental capacity who was involved, either directly or indirectly, in a deceased organ donation conversation or decision of a Welsh resident who died in a hospital in England after 1 December 2015 and was managed by the NHSBT organ donation teams covering North or South Wales.
► Any close friends of the deceased person who may want to share their perspectives on the donation process and outcome.
► Any NHSBT SNODs and managers covering Wales.

### Exclusion criteria for 'family' participant recruitment
► Under 16 years.

- ► Lacking mental capacity.
- ► Potential donor was a Welsh resident over 18 years who died in an English hospital not covered by participating SNODs.

Phase III: primary study with qualitative 1:1 or small or focus group interviews with NHSBT organ donation teams covering Wales (SNODs and managers in our coproductive project) to contextualise potential donor family decision making, reactions and responses to the Act.

SNODs and their managers will be invited by letter with accompanying study information to participate in 1:1 or small or focus group interviews at the end of the study to contextualise the findings. Interviews will be at the end of the study and last approximately 60 min.

Phase IV: comparative analysis and overarching synthesis of stages 1–3.

Study data will be analysed and findings contextualised with descriptive numerical data and additional narrative data shared by NHSBT and Welsh Government (see online supplementary appendix 1 for a summary of data sources).

## Data analysis
### Narrative textual data
With consent, interviews will be digitally recorded and transcribed verbatim in the original language. The framework approach for analysis of applied policy research will be used for all narrative textual data from questionnaires, focus groups and interviews.[22] We will use NVivo software V.11[23] to facilitate the framework analysis.[22] First, the initial five available verbatim free text in questionnaires and interview transcripts will be read and reread and key themes and categories will be identified. The definitions and boundaries of each of the emerging themes for each type of evidence from questionnaires, focus groups and interviews will then be discussed to see how these can be developed to form an a priori framework tailored for either questionnaires or transcripts in Welsh and English. Searching for additional themes will continue until all text from questionnaires and interview transcripts have been analysed and no new themes are discerned. Following analysis of Welsh language text, themes and relevant quotes will then be translated into English and back translated.

The final themes and their dimensions for text from questionnaires and interviews will then be further refined and used as the basis of charts (or matrices), which allows for themes to be compared and displayed for questionnaires and interviews, and for variations and deviant cases to be highlighted within each dataset. These charts will be overlaid with key information to preserve the original context. Second, these charts will undergo several revisions and further refinements, in an iterative process moving between the charts and the themes identified from questionnaires and interviews, until it is possible to synthesise the key findings across the datasets in a set of overall themes or categories. This stage will involve what is sometimes called the translation of themes from one data source to another. In the process of comparing the themes, we will look for explicit differences in relation to a range of factors that impact on decision making including gender, relationship of the person to the deceased, age, ethnicity and whether consent for donation was given or not and whether registration as an organ donor was viewed positively or negatively.

### Categorical questionnaire data
Completed form B and C will contain structured categorical options (such as yes, no, uncertain), which will be collated in SPSS V.22 and analysed using descriptive statistics.[24] Results will be displayed as numbers and percentages.

### Comparative analysis and overarching synthesis
We will use Oliver's synthesis framework for juxtaposing evidence across phases I–III with Welsh Government omnibus surveys and contemporaneous and previous NHSBT activity reports listed in online supplementary appendix 1.[25] We will organise data by donation decision (opt in opt out on organ donation register; expressed decision, deemed consent) mapped against whether families supported the donation decision and why. We will layer the descriptive numerical and narrative findings onto the framework to synthesise findings across the different types of evidence, working within each of the spheres of influence (the patient decision, family, NHSBT, NHS and clinical care, the law, the media campaign, previous comparative data, etc). Juxtaposing different numerical, narrative and temporal evidence in this way on the same phenomenon of interest will enable us to look for patterns, explanations, mechanisms and disconfirming cases.

As there is not a specific reporting guideline for mixed-method studies, we will draw on new guidance for reporting mixed-method syntheses[26] and the Consolidated Criteria for Reporting Qualitative Research guidelines.[27]

## Ethics and dissemination
This protocol was approved on 23 October 2015 by NHSBT Research, Innovation and Technology Advisory Group. The study was approved by the Wales Research Ethics Committee 5 NHS research ethics committee (IRAS number 190066; Rec Reference 15/WA/0414 on 25 November 2015) and the NHSBT Research and Development Committee (NHSBT ID: AP-15–02 on 24 November 2015).

The design and methods are informed by an ethical framework developed by the UK-based researchers for undertaking research with family members who are approached about organ donation and draws on the experiences of researchers working with the bereaved.[28–36] Independent governance will be provided by a steering group.

A key component of the ethical nature of the study will be the professional development and training

elements to support SNODs and research officers to conduct the study in a respectful and sensitive way. We will dovetail the bereavement support offered by researchers to participants with the bereavement services offered by intensive care units in NHS Health Boards where potential organ donors are cared for with their families, and information provided on bereavement support services shared by SNODs in NHSBT teams covering Wales. In addition, in any contact with participants we will share a bilingual information leaflet on CRUSE Bereavement Care Cymru, in case families would prefer to access free support and counselling outside of a NHS context. In appreciation of their support, research team members will plan a fund-raising activity during the study to make a donation to Cruse Bereavement Care Cymru.

Additional information concerning the specific ethical and data protection[37] issues, proposed strategies and data sharing agreement can be found in (online supplementary file 5).

## Patient and public involvement

Prior to commencement of the study, contextual baseline engagement with the public has consisted of six discussion groups and seven face-to-face interviews involving 52 participants. This contextual work was undertaken by the Welsh Government.[19] Each group was recruited to include a mix of people in terms of awareness of the NHS Organ Donor Register and included some who had joined the Register and/or carried a donor card. Black and Minority Ethnic people formed part of the sample and included Pakistani, African Caribbean, Nigerian and Chinese participants. Each group contained a mix of men and women and the sample was broadly stratified by age and socioeconomic grouping. Two groups were conducted in the Welsh language. In addition, 1006 members of the public responded to a baseline Welsh Omnibus attitudinal survey.[8] Patient and public involvement representatives were involved in prioritising the question and in deciding to fund the study. The leading charities for supporting deceased organ donor families and people with kidney failure requiring a transplant have helped shape the design and advised on appropriate methods of data collection.

The Welsh Government hosted a conference in September 2015 involving those affected by deceased organ donation and healthcare professionals involved in the donation process to explore the implementation and implications of the Act from different perspectives and to explore how best to evaluate the Act and what outcomes from different perspectives are important. These perspectives have been incorporated into the study design. PPI will continue during the study through to dissemination.[38 39]

## Building research capacity

A secondary aim of the study is to increase the confidence and capacity of NHSBT and PPI representatives to collaborate in future studies in this field.

In following Prudent healthcare principles,[39] we will use a coproduction approach, which means that the research team will work as equal partners and in collaboration with NHSBT who have a remit to support relevant research activity, and with a range of key professional stakeholders and PPI representatives to conduct the study. The coproduction element is critical to the success of the study and will involve a strong research training and capacity building component for NHSBT teams and PPI representatives working in Wales. We have worked closely with policy and clinical leads from Welsh Government and NHSBT to ensure that the proposed coproductive methods of data collection and participant recruitment are feasible, sensitive to the needs of potential donor families and NHS staff and fulfil the high ethical and data protection requirements for data sharing between two organisations.

Three development opportunities will bring NHSBT staff and PPI representatives together. At the beginning of the study, we will facilitate professional development meetings with SNODs and managers to design the data collection tools. At the end of the first year of data collection, we will present initial findings at a professional development meeting with collaborating staff from NHSBT, clinical co-applicants, policy makers and PPI representatives to see what shared learning could be used to further enhance practice development and support study data collection. We will facilitate another meeting at the end of the study to present key findings.

## Expected outcomes

The most important outcome will be a research-informed and clearer, shared understanding of deceased donor consent decisions, and in particular the reasons why people continue to refuse to support consent in a 'soft opt-out' system, to feed back into further policy and practice development. In addition, staff in NHSBT covering Wales and PPI representatives will have developed additional confidence and research capacity and capability to undertake further and equally challenging studies.

## Impact and dissemination

The study has potential for high impact as success of this Prudent health policy is dependent on the people of Wales engaging with the principles of deemed consent and donor registration and honouring the deceased person's donation decision. If sufficient people agree and change their behaviour to favour the principles of the 'soft opt-out', then the policy will likely realise the anticipated benefits for patients. If sufficient people disagree then nothing will change and the anticipated increased number of patients who benefit from cost-effective transplants will not be realised.

Understanding why people do not register on the organ donor register or why family/appointed representatives still contest the decision to donate made by the deceased person will have an impact on the design of

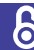

future interventions to improve organ and tissue donation rates in Wales.

The main mechanism of dissemination, knowledge transfer and maximising impact is through the uptake of project outputs by policy makers, clinicians and the public through a coproductive continuous quality improvement approach in line with Prudent healthcare principles.[15 16] There are key elements known to affect the resources required for managing successful coproductive dissemination processes and these elements are built into the project design, including:

► motivating change: creating readiness for change and overcoming resistance;
► creating a vision: mission, valued outcomes and conditions, midpoint goals and feedback;
► developing political support: assessing change agent power, identifying key stakeholders, influencing stakeholders;
► feeding back findings and jointly determining their meaning for various stakeholders;
► sustaining momentum: providing resources for professional development and research capacity building, building support systems for change agents, developing new competencies and skills, reinforcing new behaviours.[40]

Other effective elements of knowledge transfer include publication of research results in leading journals, and presentations at local, national and international conferences in the field. The research team has already demonstrated a high-quality publication record, and will continue to do so, adopting an open-access policy. We will also produce bilingual lay summaries.

**Author affiliations**
[1]School of Social Sciences, Bangor University, Wales, UK
[2]Major Health Conditions Policy Team, Directorate of Health Policy, Health and Social Services Group, Welsh Government, Cardiff, UK
[3]Department of Organ Donation, Unit 3 Cae Gwyrdd, NHS Blood and Transplant, Greenmeadow Springs Business Park, Cardiff, UK
[4]NHS Blood and Transplant, North West Regional Office, Liverpool, UK
[5]Department of Nephrology and Transplantation, Cardiff and Vale University Health Board, University Hospital of Wales, Cardiff, UK

**Twitter** @OrganDonation_

**Acknowledgements** Fiona Wellington: Head of Operations NHSBT for supporting the study. Christian Brailsford: NHSBT provided advice and support to agree a mutual data sharing agreement and negotiate NHS ethics and NHSBT RINTAG and NHSBT R&D processes. Pat Vernon (Policy Lead Welsh Government), Ian Jones (Research and Evaluation Lead), Caroline Lewis (Organ Donation Policy Manager) provided a Government perspective and shared research carried out prior to implementation of the Act. Donald Fraser: Lead of the Wales Kidney Research Unit supported development of the funding application and serves as independent Chair of the steering group. Jo Mitchell: research support officer. North West NHSBT Team: Ben Armstrong, Adam Barley Angela Campion-Sheen, Laura Ellis-Morgan, Rebecca Gallagher, Sharon Hallam, Phil Jones, Andrew Mawson, Abi Roberts, Tracey Rhodes, Helen Bullock, Andrea Jones, Kathryn Alletson, Jane Monks, Emma Thirlwall, Dawn Lee, Nicky Hargreaves, Lisa Welsh, Gill Drisma, Sue Duncalf. South Wales NHSBT Team: Angharad Griffiths, Lucy Barnes, Charlotte Goodwin, Guy Heathcote, Gail Melvin, Michael Tobin, Lisa Morgan, Nicola Newbound, Michelle Powell, Stephen Regan, Fiona Rogers, Susie Cambray, Kathy Rumbleow, Lynne Woolcocks, Janet Woodley, Beth Moss, Louise Colson. NHS staff: SianGriffin (Consultant Nephrologist, Department of Nephrology and Transplantation), Katja Empson, Sam Sandow, Carl Stephenson (Clinical Leads Organ Donation), Francesca Stevens (Tissue Services NHSBT), Maggie Stratton (PR Officer NHSBT). Jeanette Bourne and CRUSE Bereavement Care, who provided leaflets signposting bereavement support for participants. Patient and public representatives: Sarah Thomas, Janet Thickpenny, Gethin Rhys, Michael Rhys, Maria Mesa, Roon Adams, Michael and Jess Houlston, Maria Battle, Anna Bates for providing guidance and advice on the focus of the study. Janet Williams and Gloria Owen for providing advice and feedback on processes, participant facing documents and the detailed funding application which contained this protocol. Patient and public representative organisations: Flintshire Deaf Association, Llanelli Multicultural Network, Big Lottery, Churches Together in Wales, Women Connect First, BAWSO, Race Equality First, Believe, Donor Family Network, Rita's Café for helping to set up the patient and public network and supported the study proposal. Gareth Wyn Roberts: Consultant Nephrologist, Cardiff and Vale University Health Board provided detailed advice on clinical processes and the new Act. Catherine Robinson: Former Head of School of Social Sciences, Bangor University, supported submission of the funding application following high-level discussions and commented on a section of the application.

**Contributors** JN Chief Investigator conceptualised the idea, put the team together, designed the study and procedures and drafted the protocol.MS Consultant Transplant and Organ Retrieval Surgeon, Clinical Lead for Transplantation, Cardiff and Vale Health Board, advised on key research team members and stakeholders to bring into the research team, proposed changes in the law and key research questions to address. KM Formerly Regional Manager South Wales and South West, NHSBT and now Major Health Conditions Policy Team, Directorate of Health Policy, Health and Social Services Group, Welsh Government, advised on key changes to policy and practice, study design and processes, data collection tools and implementation of the study. PW Regional Manager South Wales NHSBT advised on changes to policy and practice, study design and processes, data collection tools and implementation of the study. AR Specialist Nurse in Organ Donaiton NHSBT advised on the role of the Specialist Nurse in organ donation, study design and processes, data collection tools and implementation of the study. LM Research Officer, finalised study procedures and data collection processes, designed the study documentation and logos and supported production of applications to the NHS REC and NHSBT R&D committees. All authors approved the final manuscript.

**Funding** The study was funded by Health and Care Research Wales. Project Reference 1129.

**Competing interests** None declared.

**Ethics approval** Wales NHS Research Ethics Committee 5.

**Provenance and peer review** Not commissioned; externally peer reviewed.

**Data sharing statement** The protocol contains information on the data sharing agreement between NHSBT and Bangor University (Supplemental File. Appendix 5).

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
