## [Reviewer comments · BMJ Open]

ARTICLE DETAILS

TITLE (PROVISIONAL)	Family attitudes, actions, decisions and experiences following implementation of deemed consent and the Human Transplantation (Wales) Act 2013: Mixed-method study protocol
AUTHORS	Noyes, Jane ; Morgan, Karen; Walton, Philip; Roberts, Abigail; Mc Laughlin, Leah; Stephens, Micheal

VERSION 1 – REVIEW

REVIEWER	Nigel Monaghan Public Health Wales Wales
REVIEW RETURNED	04-May-2017

GENERAL COMMENTS	Reference could be made to NICE guidance and key role of specialist organ donation nurses https://www.nice.org.uk/guidance/cg135/evidence/full-guideline-184994893 May be worth highlighting up front some potential limitations of individual components of this study under strengths and limitations, and how the mixed methods approach assists in addressing these potential limitations. Minor typographical errors as files was presented on ScholarOne include: Page 9 lines 34/35 unneeded letter "t" Page 10 lines 36/37 superscript needed for references "13-14" Presumably the write up of findings will explore the complex changes at play e.g.: Changes in legal presumption of consent Changes in processes for organ donation staff Changed role of relatives in this process Impact on relatives Impact on organ donation rate
--

REVIEWER	ALVIN LI OTTAWA HOSPITAL RESEARCH INSTITUTE CANADA
REVIEW RETURNED	04-May-2017

GENERAL COMMENTS	This manuscript describes a large-scale mixed-methods evaluation of the impact of the Human Transplantation Act 2013. The study protocol has many strengths such as including patients and policy-makers. The study has a great description of the Act which will be of interest to many readers. Suggestions: Introduction: Could the authors briefly describe the difference between 'soft opt out' and 'opt-out/presumed consent'? Could the authors briefly describe the demographics of Wales? (e.g. population size) Also, could they provide some statistics of Wales' registration rate, opt-out rate, methods of registration/opt-out etc? Methods Has any pilot testing been done? If yes, please describe Data Analysis It seems that most of the analysis will be qualitative. Could you further clarify how this is a mixed-method study? Is there an interview guide? You state that you will look for explicit differences in relation to gender, age, ethnicity - how is ethnicity being recorded? It is not on the questionnaires. Are you using any software for the qualitative analysis? Other: Did the patient representatives contribute to the writing or review of the manuscript? Could you comment if you will be using any reporting guidelines for the writing of your end of study publication?
---

REVIEWER	Stuart Nicholls Children's Hospital of Eastern Ontario Research Institute
REVIEW RETURNED	16-May-2017

GENERAL COMMENTS	Thank you for the review of this study protocol. As this is a protocol, my comments are limited, but I do have some specific comments:  1. In the abstract the formatting of dates should be consistent (1.12.15 and 1/12/15) Introduction  2. To what extent is the goal actually behaviour change. The Act may in fact increase donation rates with absolutely no change in behaviour (no increase in opt in but also no increase in active
--

decisions not to donate). In fact, the default position is changed is that of the system.

3. I'm wondering what 'neutral' means in this context. If the whole series of changes are set up to change practice, how is the media campaign neutral - especially when the targeted goal is to increase donation rates to 80% and improve known decisions. I doubt that a media campaign that resulted in no increased donation rates would be deemed a 'success'.

Methods

1. Phase 1 is unclear. Given the 18 month retrospective window, will these families still be in touch with the SNODs? Have these forms been routinely completed since 1st December 2015?

A major limitation of this phase is that there is no data prior to 1 December 2015. As such, the study is incapable to determine the impact on donor responses as there is no prior data for comparison. As such, the best that can be hoped for is the analysis of data from 1 December 2015. Importantly, this will miss the important 2 years of media campaigns and effect that this has achieved. Indeed, in reality the study requires 3 time periods to address the question: (a) prior to the implementation of the act and start of the media campaign; (b) the period following the act and during the implementation of the media campaign (December 2013-1 December 2015; (c) after 1 December 2015. Indeed, this would allow for something akin to an interrupted time series analysis to look at whether there were changes in trends over time.

Secondly, there is no detail of the planned analysis of the Form B data. How is this to be used, what will be analyses, what outcomes are being looked at? This really needs much more information.

4. For Phase 2 – how will families be identified. How are participants selected? Upon what grounds are they chose? Is it random sampling or purposive sampling? Will you explicitly seek out participants who declined donation?

5. As laid out Phase 3 includes the Inclusion Criteria for 'Family' participant recruitment, yet phase 3 is for SNODs and managers.

6. The organisation is a little confusing and I would suggest laying out each phase in detail with methods and analyses grouped by phases, rather than laying out the phases and having a larger analysis section. At present it is hard to align the methods and analyses for each phase.

7. In the data analysis section the authors refer to the "Framework approach for analysis of applied policy research". It is not clear what the framework is – or how it can be applied. There really needs to be much more exposition of the analysis process for all sections.

8. Equally, the analysis of Form B is uninformative as stated. What is going to be analysed (primary outcome of interest). What is the question to be answered?

9. What processes are in place for the translation of bilingual materials – is there a process of back translation to ensure accuracy?

10. What is Oliver's approach for juxtaposing evidence? More needs to be said about this approach.

11. More could be said about the findings from the prior patient and public involvement at the outset. This would help frame the current study.

12. The expected outcomes seem to have limited overlap with the evaluation of policy. Why people continue to refuse to support consent could be undertaken through a purposive sampling of declining families – and much research has been done on this. If why people decline relates to issues that are not amenable to legislative change or practice change on the part of the

	professionals, is that a failure of policy? There seems to be a misalignment between the expected outcomes and the promulgated goals. 13. Indeed, readiness to change and other provider behaviours suggest providers need to change. Has this been shown. It may also be worth considering the use of the Theoretical Domains Framework (TDF) to help guide interview content if it is felt that there could be barriers to change in order to identify potential issues for future intervention.
--	--

VERSION 1 – AUTHOR RESPONSE

Reviewer 1.

Reference could be made to NICE guidance and key role of specialist organ donation nurses
<https://www.nice.org.uk/guidance/cg135/evidence/full-guideline-184994893>

May be worth highlighting up front some potential limitations of individual components of this study under strengths and limitations, and how the mixed methods approach assists in addressing these potential limitations.

Minor typographical errors as files was presented on ScholarOne include:

Page 9 lines 34/35 unneeded letter "t"

Page 10 lines 36/37 superscript needed for references "13-14"

Presumably the write up of findings will explore the complex changes at play e.g.:

Changes in legal presumption of consent

Changes in processes for organ donation staff Changed role of relatives in this process Impact on relatives Impact on organ donation rate

Thank you for flagging the NICE Guidance – we agree that the reference will be a helpful addition.

Yes agree- the Editor has asked us to add a bullet point outlining a potential limitation. We have also expanded the overarching synthesis section to explain how these mixed-method data will be brought together and for what purpose.

Thank you for picking up these typos. We have corrected them.

Yes – these are exactly the things that we will explore when addressing the aim of ‘what happened’ when the new law was implemented.

Reviewer 2

This manuscript describes a large-scale mixed-methods evaluation of the impact of the Human Transplantation Act 2013. The study protocol has many strengths such as including patients and policy-makers. The study has a great description of the Act which will be of interest to many readers.

Suggestions:

Introduction:

Could the authors briefly describe the difference between 'soft opt out' and 'opt-out/presumed consent'?

Could the authors briefly describe the demographics of Wales? (e.g. population size) Also, could they provide some statistics of Wales' registration rate, opt-out rate, methods of registration/opt-out etc?

Methods

Has any pilot testing been done? If yes, please describe

Data Analysis

It seems that most of the analysis will be qualitative. Could you further clarify how this is a mixed-method study?

Is there an interview guide?

You state that you will look for explicit differences in relation to gender, age, ethnicity - how is ethnicity being recorded? It is not on the questionnaires.

Are you using any software for the qualitative analysis?

Other:

Did the patient representatives contribute to the writing or review of the manuscript?

Could you comment if you will be using any reporting guidelines for the writing of your end of study publication?

Thank you for this positive comment.

We have added a clarification in the introduction.

We have added some additional details in the introduction about the demographics of Wales. Opt in and out rates from the Organ Donation Register form part of the evaluation and will be reported in the full study report.

The new questionnaires and interview schedules were developed with NHSBT staff and a Welsh Government Civil Servant to ensure that the appropriate questions were asked and data captured. There was no formal piloting as there was no time, but the forms were found to be fit for purpose.

The research team are collecting additional data to supplement the numerical and narrative data collected by Welsh Government and NHSBT. We have included a summary of the additional routinely or specifically collected data that we will include in the analysis (Appendix 1).

Yes – we have included the participant interview guide as an additional online file (Appendix 4).

Thank you for raising this point. We are only collecting additional data not already collected by NHSBT. As described above, we have included a summary of the routinely or specially collected data to include in the analysis, which covers these issues.

Yes – we have added a note to indicate that we opted to use NVIVO.

Yes – this protocol is taken directly from the original funding application, which had PPI input. PPIs are acknowledged in the relevant section.

There are no specific reporting guidelines for mixed-method studies. One author on this protocol is currently contributing to writing such guidelines and the article is currently in press with the Journal of Clinical Epidemiology. We have added a citation to this in press article. Where appropriate we will also refer to COREQ - the reporting guidelines for qualitative studies. A note stating this has been added to the protocol.

Reviewer 3

Thank you for the review of this study protocol. As this is a protocol, my comments are limited, but I do have some specific comments:

Thank you for your comments.

1. In the abstract the formatting of dates should be consistent (1.12.15 and 1/12/15)

Thank you for pointing out this discrepancy. Corrected.

Introduction

2. To what extent is the goal actually behaviour change. The Act may in fact increase donation rates with absolutely no change in behaviour (no increase in opt in but also no increase in active decisions not to donate). In fact, the default position is changed is that of the system.

3. I'm wondering what 'neutral' means in this context. If the whole series of changes are set up to change practice, how is the media campaign neutral - especially when the targeted goal is to increase donation rates to 80% and improve known decisions. I doubt that a media campaign that resulted in no increased donation rates would be deemed a 'success'.

In a protocol, the programme theory (how the intervention was intended to work) is presented from the perspective of the intervention designer (ie. the Welsh Government). In this type of evaluation we will look at 'what happened' when the intervention was implemented and whether it worked as intended. We will be looking to answer the questions raised by the reviewer as part of the evaluation. Nonetheless, 'Doing nothing' is also considered a 'behaviour' under the Act and 'doing nothing' is an option presented to the population of Wales. If people 'do nothing' then it is assumed that they have no objection to organ donation. The relatives/appointed representatives still have to behave in a specific way for consent rates to increase. We have documented these anticipated intended behaviours in the protocol under the section 'how the intervention is intended to work'. In this evaluation we will be looking to see what happened and whether potential donors and their relatives/appointed representatives behaved as anticipated or not, or indeed whether the intervention was indeed a behavioural change intervention as originally conceived.

Thank you for this observation. The independent evaluation team played no role in designing the intervention and media campaign – we are only evaluating it. In a protocol, the programme theory (how the intervention was intended to work) is presented from the perspective of the intervention designer (ie. the Government). The Government have presented the media campaign as neutral so

this is the starting point of the evaluation team. The evaluation team will have an opportunity to comment on this issue of neutrality when the evaluation is completed.

Methods

1. Phase 1 is unclear. Given the 18 month retrospective window, will these families still be in touch with the SNODs? Have these forms been routinely completed since 1st December 2015?

Thank you for highlighting this query. In the protocol we explain the following that addresses your query:

'SNODs will complete an anonymised electronic 1 page form [Appendix.1: Form B] for every approach conversation that will be filled out as soon as possible after they have disengaged from speaking with the potential donor family. No participant or patient identifiable information will be recorded. SNODs will complete the form (electronic or paper) using information gathered from their routine conversations with potential donor families.'

A major limitation of this phase is that there is no data prior to 1 December 2015. As such, the study is incapable to determine the impact on donor responses as there is no prior data for comparison. As such, the best that can be hoped for is the analysis of data from 1 December 2015. Importantly, this will miss the important 2 years of media campaigns and effect that this has achieved. Indeed, in reality the study requires 3 time periods to address the question: (a) prior to the implementation of the act and start of the media campaign; (b) the period following the act and during the implementation of the media campaign (December 2013-1 December 2015; (c) after 1 December 2015. Indeed, this would allow for something akin to an interrupted time series analysis to look at whether there were changes in trends over time.

Secondly, there is no detail of the planned analysis of the Form B data. How is this to be used, what will be analysed, what outcomes are being looked at? This really needs much more information.

This was a most useful comment - thank you - as it highlighted a patch and paste error, which gave the impression that there was no retrospective comparison data, when in fact there is. We have corrected the typo in the manuscript and to reassure the referee, we do intend to make comparisons with what happened in previous years with routinely collected NHSBT data. We already cite some key data on retrospective consent rates (going back to 2012) in the introduction as context and also state that we will use routinely collected NHSBT data to contextualise the study findings. Given that the international audience will not be familiar with these additional data, we have also included a table of data sources that we will be drawing on (Appendix 1).

Thank you for pointing this out. Of note – all retrospective data is collected from 1st April to 31st March and these dates do not map exactly onto the key time periods that the reviewer suggests so we will be looking for overall temporal trends.

We have added additional clarification concerning the simple descriptive analysis of Forms B and C in the analysis section and readers can locate Forms B and C in additional files.

4. For Phase 2 – how will families be identified. How are participants selected? Upon what grounds are they chose? Is it random sampling or purposive sampling? Will you explicitly seek out participants who declined donation?

The recruitment strategy is designed so that study information will be shared with family members of all potential donors over an 18m period. So in answer to your question - yes we will seek to recruit participants who declined donation. We will interview family members from a minimum of 50 cases in order to obtain maximum variation. Multiple purposive recruitment strategies will be used, which will be constantly reappraised to ensure that the sample meets the needs of the study. The strategies are detailed in the protocol:

'We will use a range of methods to recruit participants that are sensitive and individually tailored. SNODs will use their discretion and knowledge of the family to select the most appropriate options and times to share information about the study with families/appointed representatives. Recruitment options include via direct contact with families by SNODs (with the option to using consent to contact form), and sharing study information in person; and by sending out a study invitation with information attached to routine follow-up communication by NHSBT; by direct mailing of study invitation and information by NHSBT; via adverts in the media, and through snowball sampling'

5.As laid out Phase 3 includes the Inclusion Criteria for 'Family' participant recruitment, yet phase 3 is for SNODs and managers.

Thank you for pointing this out. The current structure mirrors the suggested journal template, but we agree that this text is better placed elsewhere so the text has been moved to achieve a better reader experience.

6.The organisation is a little confusing and I would suggest laying out each phase in detail with methods and analyses grouped by phases, rather than laying out the phases and having a larger analysis section. At present it is hard to align the methods and analyses for each phase.

Thank you for this suggestion. We have described the analysis methods by type of data rather than phase otherwise we would be repeating ourselves as similar methods of analysis are used across the phases. We have however relabelled the subheadings in the analysis section to make it easier for the reader to navigate.

7.In the data analysis section the authors refer to the "Framework approach for analysis of applied policy research". It is not clear what the framework is – or how it can be applied. There really needs to be much more exposition of the analysis process for all sections.

We explain the application of the Framework approach in two long paragraphs on page 18.

8. Equally, the analysis of Form B is uninformative as stated. What is going to be analysed (primary outcome of interest). What is the question to be answered?

This type of evaluation sets out to establish 'what happened'. The research questions and objectives are described on page 14. We have also restructured the analysis section to make the analysis of questionnaire data clearer.

9. What processes are in place for the translation of bilingual materials – is there a process of back translation to ensure accuracy?

Wales is a bilingual country and Welsh and English have equal status in Law. The research team are bilingual in Welsh and English and can forward and back translate with ease – this is normal practice in Wales.

10. What is Oliver's approach for juxtaposing evidence? More needs to be said about this approach.

We have added more detail to this section and agree that it helps with clarity.

11. More could be said about the findings from the prior patient and public involvement at the outset. This would help frame the current study.

The current manuscript is already over the journal word limit and each of the referees has requested more details in key areas. There is no space to expand on the findings of the public consultations that influenced the drafting of the Act. We have however included all the references so that readers can read these extensive consultations at their leisure.

12. The expected outcomes seem to have limited overlap with the evaluation of policy. Why people continue to refuse to support consent could be undertaken through a purposive sampling of declining families – and much research has been done on this. If why people decline relates to issues that are not amenable to legislative change or practice change on the part of the professionals, is that a failure of policy? There seems to be a misalignment between the expected outcomes and the promulgated goals.

A systematic review was previously commissioned by Welsh Government to determine the reasons why people supported or declined to support organ donation in different systems. This review is listed in Appendix 1 and we will update it as part of the secondary background work to contextualise study findings.

We will be evaluating policy, the intervention and practice and the purpose is to establish if the policy (ie the intervention) and associated intervention strategies worked as intended, what happened when implemented and why, and to determine the mechanisms (which may or may not be due to the policy).

13. Indeed, readiness to change and other provider behaviours suggest providers need to change. Has this been shown.

It may also be worth considering the use of the Theoretical Domains Framework (TDF) to help guide interview content if it is felt that there could be barriers to change in order to identify potential issues for future intervention.

In looking at 'what happened' we will be exploring the experiences of potential/actual donor families of the health system as well as the organ donation process. Thank you for suggesting the Theoretical Domains Framework. We are very familiar with this framework, which focuses on professional provider domains and perspectives. As the focus of our new data collection is primarily on the perspectives of lay family members, close friends and appointed representatives who are approached about organ donation and their loved one, we do not feel that it is the best fit to frame the analysis. Nonetheless, thank you for the suggestion.

In addition, Welsh Government commissioned a series of sequential focus groups with Specialist Nurses in Organ Donation and other key clinical staff to ascertain their attitudes and experiences to the changes (see appendix 1). These data have already been analysed by another team and findings have been shared with our study team. These data will be used to contextualise our findings.

VERSION 2 - REVIEW

REVIEWER	Alvin Li Ottawa Hospital Research Institute
REVIEW RETURNED	31-Jul-2017

GENERAL COMMENTS	RE: comments about Wale's registration rates - you mention that opt in and out rates will be reported in the full study report. however, is there no historical data on this?
---